# Neuroendocrine gene expression coupling of interoceptive bacterial food cues to foraging behavior of *C. elegans*

Sonia A Boor[1,2], Joshua D Meisel[2,3], Dennis H Kim[1]*

[1]Division of Infectious Diseases, Department of Pediatrics, Boston Children's Hospital and Harvard Medical School, Boston, United States; [2]Department of Biology, Massachusetts Institute of Technology, Cambridge, United States; [3]Department of Molecular Biology, Massachusetts General Hospital, Boston, United States

*For correspondence:
dennis.kim@childrens.harvard.edu

Competing interest: The authors declare that no competing interests exist.

**Abstract** Animal internal state is modulated by nutrient intake, resulting in behavioral responses to changing food conditions. The neural mechanisms by which internal states are generated and maintained are not well understood. Here, we show that in the nematode *Caenorhabditis elegans*, distinct cues from bacterial food – interoceptive signals from the ingestion of bacteria and gustatory molecules sensed from nearby bacteria – act antagonistically on the expression of the neuroendocrine TGF-beta ligand DAF-7 from the ASJ pair of sensory neurons to modulate foraging behavior. A positive-feedback loop dependent on the expression of *daf-7* from the ASJ neurons acts to promote transitions between roaming and dwelling foraging states and influence the persistence of roaming states. SCD-2, the *C. elegans* ortholog of mammalian anaplastic lymphoma kinase (ALK), which has been implicated in the central control of metabolism of mammals, functions in the AIA interneurons to regulate foraging behavior and cell-non-autonomously control the expression of DAF-7 from the ASJ neurons. Our data establish how a dynamic neuroendocrine *daf-7* expression feedback loop regulated by SCD-2 functions to couple sensing and ingestion of bacterial food to foraging behavior. We further suggest that this neuroendocrine feedback loop underlies previously characterized exploratory behaviors in *C. elegans*. Our data suggest that the expression of *daf-7* from the ASJ neurons contributes to and is correlated with an internal state of 'unmet need' that regulates exploratory foraging behavior in response to bacterial cues in diverse physiological contexts.

## eLife assessment

This **important** manuscript focuses on the mechanisms by which food signals and food ingestion modulate animal foraging. The authors provide **convincing** support for the interesting idea that chemosensory and interoceptive signals converge on transcriptional regulation of the TGF-beta ligand DAF-7 in a single pair of *C. elegans* chemosensory neurons (ASJ) to regulate behavior. Their studies implicate a conserved signaling molecule, ALK, in this regulation, suggesting a conserved link between food cues and the neuroendocrine control of foraging behavior.

## Introduction

Internal states, such as fear, arousal, and hunger, are shaped by the integration of information about internal and external conditions and can result in the modulation of various physiological and behavioral outputs (*Flavell et al., 2022*). As animals encounter different food environments, information about internal nutritional status and external food quality can elicit transitions between internal states that either favor the exploration of new places or exploitation of the current environment. Many

animals, including mammals, zebrafish, *Drosophila melanogaster*, and the roundworm *Caenorhabditis elegans,* increase exploration in response to food deprivation in order to increase chances of a food encounter (*Ben Arous et al., 2009*; *Connolly, 1966*; *Gutman et al., 2007*; *Herbers, 1981*; *Johnson et al., 2020*; *Overton and Williams, 2004*; *Russell et al., 1987*). However, the mechanisms behind how animals couple changes in food availability to internal states that influence foraging behavior are not well understood. Insight into the cellular and organismal mechanisms governing internal states may enhance understanding of the dysregulation of internal states that is thought to contribute to many human psychiatric and neurological diseases (*Flavell et al., 2022*; *Yap and Greenberg, 2018*).

*C. elegans* forage for microbes that grow on decaying organic matter, where they encounter fluctuations in not only the quantity of nutritious bacterial food available but also in the quality and pathogenicity of this bacterial food (*Kim and Flavell, 2020*). As they navigate their food environments, *C. elegans* exhibit two-state foraging and feeding behavior known as roaming and dwelling (*Ben Arous et al., 2009*; *Flavell et al., 2013*; *Flavell et al., 2020*; *Fujiwara et al., 2002*). On abundant nutritious food, animals spend about 80% of their time dwelling and 20% of their time roaming; as food becomes scarcer or lower in quality, animals increase the proportion of their time roaming (*Ben Arous et al., 2009*). The internal states that underlie *C. elegans* roaming and dwelling responses to changing food conditions present an experimentally tractable paradigm in which to study how internal states are regulated. Furthermore, the shared modulators of internal states and high degree of genetic conservation between *C. elegans* and humans suggest that understanding *C. elegans* feeding and foraging behavior could have relevant implications for human health and disease.

Aided by a complete connectome (*White et al., 1986*), a growing body of work has examined neural circuits and their effects on foraging behavior in *C. elegans* (*Ji et al., 2021*; *Pradhan et al., 2019*). The excitation or inhibition of neurons in these circuits elicits rapid behavioral modulation as well as persistent behavioral states through recurrent neuronal firing. Neuronal dynamics are additionally influenced by neuromodulators (*Bargmann, 2012*). Neuromodulators play a key role in mediating internal states by functioning at titratable levels over longer timescales and farther distances than direct synaptic signaling (*Flavell et al., 2022*; *Sengupta, 2013*). For instance, the activation of a small group of neurons can induce transitions between roaming and dwelling states through conserved serotonin and pigment dispersing factor (PDF) signaling (*Flavell et al., 2013*; *Ji et al., 2021*). Other neuromodulators such as dopamine and octopamine can further influence foraging behavior and food-dependent locomotion (*Churgin et al., 2017*; *Oranth et al., 2018*; *Sawin et al., 2000*). Activity-dependent gene expression in neurons has been established to have key roles in the development and plasticity of neuronal circuits, but less is known about how changes in neuronal gene expression may shape internal states driving behavior (*Yap and Greenberg, 2018*). As transcription occurs on a slower timescale than neuronal firing or neuromodulator release, dynamic gene expression could be important in regulating the persistence of internal states.

We have been studying the dynamic temporal and neuron-specific expression of *daf-7*, which encodes a TGF-beta ligand that is involved in the neuroendocrine regulation of a diverse range of behaviors in *C. elegans*, including the dauer developmental decision, longevity, metabolism, and feeding and foraging behavior (*Ben Arous et al., 2009*; *Greer et al., 2008*; *Ren et al., 1996*; *Schackwitz et al., 1996*; *Shaw et al., 2007*). Expression of *daf-7* is restricted to a limited set of sensory neurons, including the ASI neurons (*Meisel et al., 2014*; *Ren et al., 1996*; *Schackwitz et al., 1996*). DAF-7 expression from the ASI neurons was shown to respond to changing environmental conditions, such as crowding and food levels (*Entchev et al., 2015*; *Ren et al., 1996*; *Schackwitz et al., 1996*). Previously, we observed that highly dynamic *daf-7* expression can be observed in the ASJ neurons, with induction of expression in response to secondary metabolites produced by pathogenic *Pseudomonas aeruginosa* PA14, which is necessary for pathogen avoidance behavior (*Meisel et al., 2014*). In addition, we have shown that upon the onset of reproductive maturity, male *C. elegans* upregulate *daf-7* expression in their ASJ neurons to promote male mate-searching behavior (*Hilbert and Kim, 2017*).

Here, we have identified that the ingestion of bacterial food regulates the expression of a single gene in two neurons to shape internal state dynamics driving foraging behavior in *C. elegans*. We observed that *daf-7* transcription levels in the ASJ neurons couple foraging behavior to changes in bacterial food ingestion, under the control of the highly conserved receptor tyrosine kinase SCD-2/ALK. The relationship between gene transcription and behavioral states across organisms is largely

uncharacterized (*Yap and Greenberg, 2018*), and our results establish SCD-2/ALK-regulated dynamic *daf-7* expression as a gene expression correlate and driver of internal states underlying foraging behavior in response to changing nutritional conditions.

## Results

### Ingested food inhibits *daf-7* expression in the ASJ neurons

Our lab has previously reported that when adult hermaphrodites are fed the normal *Escherichia coli* OP50 food source, *pdaf-7::GFP* expression is restricted to the ASI chemosensory neurons (*Hilbert and Kim, 2017*; *Meisel et al., 2014*). In contrast, we observed that when animals were fed OP50 treated with aztreonam, an antibiotic that causes the bacteria to form inedible long strands (*Gruninger et al., 2008*), *pdaf-7::GFP* was expressed in both the ASI and ASJ neurons (*Figure 1A, C, and D*). These data suggested that ingestion of food inhibits *daf-7* expression in the ASJ neurons. However, we did not observe *daf-7* expression in the ASJ neurons in the complete absence of food ('Empty'), which indicated that some bacterial component of non-ingestible aztreonam-treated *E. coli* OP50 was required for the induction of *daf-7* in the ASJ neurons in the absence of ingested food (*Figure 1A, C, and D*). The non-ingestible food signal that results in the upregulation of *daf-7* expression in the ASJ neurons on aztreonam-treated food could be diffusible, volatile, or mechanosensory. To identify the nature of the non-ingestible food cue, we exposed animals to food that had been seeded on the lid of the plate (exposing animals to only the volatile food cues) ('Lid') or underneath the agar (exposing animals only to diffusible food cues) ('Bottom') in the absence of ingestible food and probed *daf-7* expression. We observed induction of *daf-7* in the ASJ neurons when OP50 is seeded under the agar of the plates, but not when present on the lid of the plate, suggesting that a bacteria-derived food signal that diffuses through the agar is necessary for the induction of *daf-7* expression in ASJ in the absence of ingested food (*Figure 1A–F*). Thus, these data suggest that in hermaphrodites feeding on *E. coli* OP50, there is an external water-soluble diffusible cue from the bacteria that stimulates *daf-7* expression from the ASJ neurons, but this expression is inhibited by a second bacteria-derived, interoceptive cue generated from the ingestion of bacteria (*Figure 1B*).

We next asked whether the inhibition of *daf-7* expression in the ASJ neurons by ingestion of bacterial food was mediated by the early detection of ingested food or the delayed nutritional effects of bacterial food consumption on organismal physiology. The kinetics of *pdaf-7::GFP* induction in the ASJ neurons when animals were moved from ingestible to non-ingestible food showed a rapid upregulation within 2–3 hr for the GFP fluorescence in the ASJ neurons to become visible (*Figure 1G*). Given the delay for GFP folding and accumulation to visible levels, these data suggested that the absence of ingested food rapidly led to the induction of *daf-7* expression in the ASJ neurons. Recent work has described the interoceptive sensing of ingested food by the acid-sensing ion channels encoded by *del-3* and *del-7*, which are expressed in the minor neurites of the NSM neurons where they detect food in the pharynx and mediate behavioral slowing upon encountering food (*Rhoades et al., 2019*). We observed that *del-3;del-7* animals have elevated expression *daf-7* in the ASJ neurons on ingestible food but show wild-type levels of *daf-7* expression on non-ingestible food or no food (*Figure 1H*). These observations are consistent with the hypothesis that the detection of an interoceptive ingested food signal in the pharynx, mediated by DEL-3 and DEL-7, inhibits *daf-7* expression in the ASJ neurons.

### *daf-7* expression in the ASJ neurons promotes roaming behavior

Next, we sought to understand how changes in *daf-7* expression in the ASJ neurons in response to changing food conditions might modulate food-dependent behavior. Feeding *C. elegans* alternate between two distinct behavioral states known as roaming and dwelling (*Ben Arous et al., 2009*; *Fujiwara et al., 2002*). On nutritious food, wild-type animals spend most of their time dwelling, a feeding state marked by a low movement speed and high body bending angle. As food conditions worsen, animals will decrease the fraction of time dwelling and increase the fraction of time roaming, a foraging state marked by high speed and low curvature (*Ben Arous et al., 2009*). Prior analysis of roaming and dwelling has identified that animals increase the proportion of time roaming on aztreonam-treated food, and genetic analysis has revealed a role for DAF-7 in promoting roaming behavior, with *daf-7* animals spending a decreased fraction of time roaming compared to wild-type animals (*Ben Arous et al., 2009*). Using both worm-tracking software to quantify roaming and dwelling (*WormLab, 2020*)

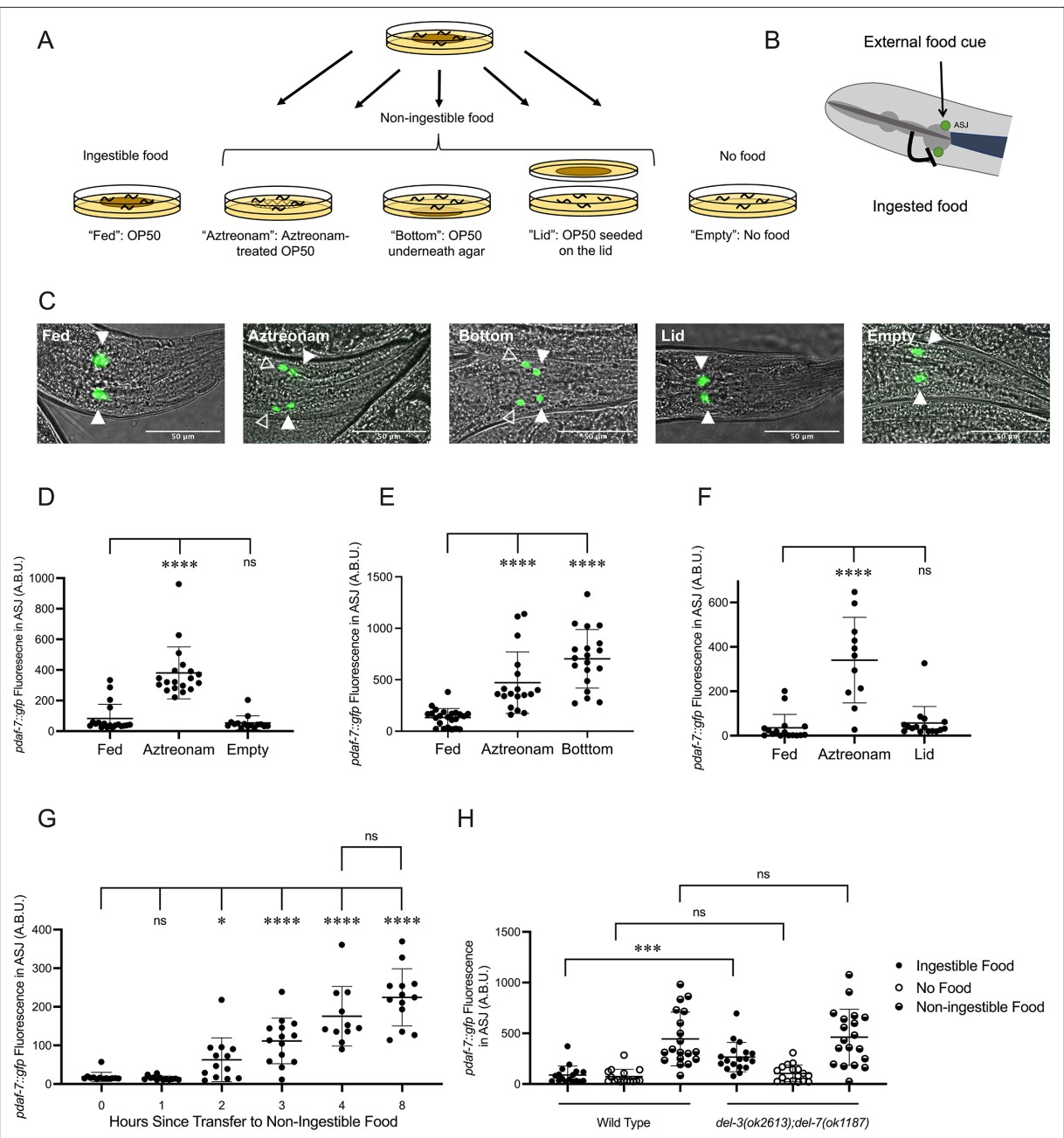

**Figure 1.** Ingestion of bacterial food inhibits *daf-7* expression in the ASJ neurons. (**A**) Schematic of experimental exposure to varied food conditions. Animals were grown to adulthood on edible *E. coli* OP50, then transferred to various experimental food conditions: 'Fed': *E. coli* OP50; 'Aztreonam': OP50 treated with aztreonam; 'Bottom': OP50 was seeded underneath the agar of the plate; 'Lid': Animals placed on agar with no food but where food was seeded on a spot of agar on the inside of the lid of the plate; 'Empty': no food. (**B**) Model for the convergence of external and ingested food signals on *daf-7* expression in the ASJ neurons. (**C**) *pdaf-7::gfp* expression pattern in animals under different food conditions, from left to right: 'Fed,' 'Aztreonam,' 'Bottom,' 'Lid,' 'Empty.' Filled triangles indicate the ASI neurons; open triangles indicate the ASJ neurons. Scale bar indicates 50 μm. (**D, E, F**), Maximum fluorescence values of *pdaf-7::gfp* in the ASJ neurons of adult animals under various food conditions. Each point represents an individual animal, and error bars indicate standard deviation. ****p<0.0001, ns, not significant as determined by an unpaired two-tailed t-test. (**G**) Maximum fluorescence values of *pdaf-7::gfp* in the ASJ neurons of adult animals at various time points after being moved from 'Fed' to 'Bottom' conditions. Each point represents an individual animal, and error bars indicate standard deviation. ****p<0.0001, *p<0.05, ns, not significant as determined by an unpaired two-tailed t-test. (**H**) Maximum fluorescence values of *pdaf-7::gfp* in the ASJ neurons of adult wild-type and *del-3(ok2613); del-7(ok1187)* under 'Fed,' 'Bottom,' and 'Empty' conditions. Each point represents an individual animal, and error bars indicate standard deviation. ***p<0.001, ns, not significant as determined by an unpaired two-tailed t-test.

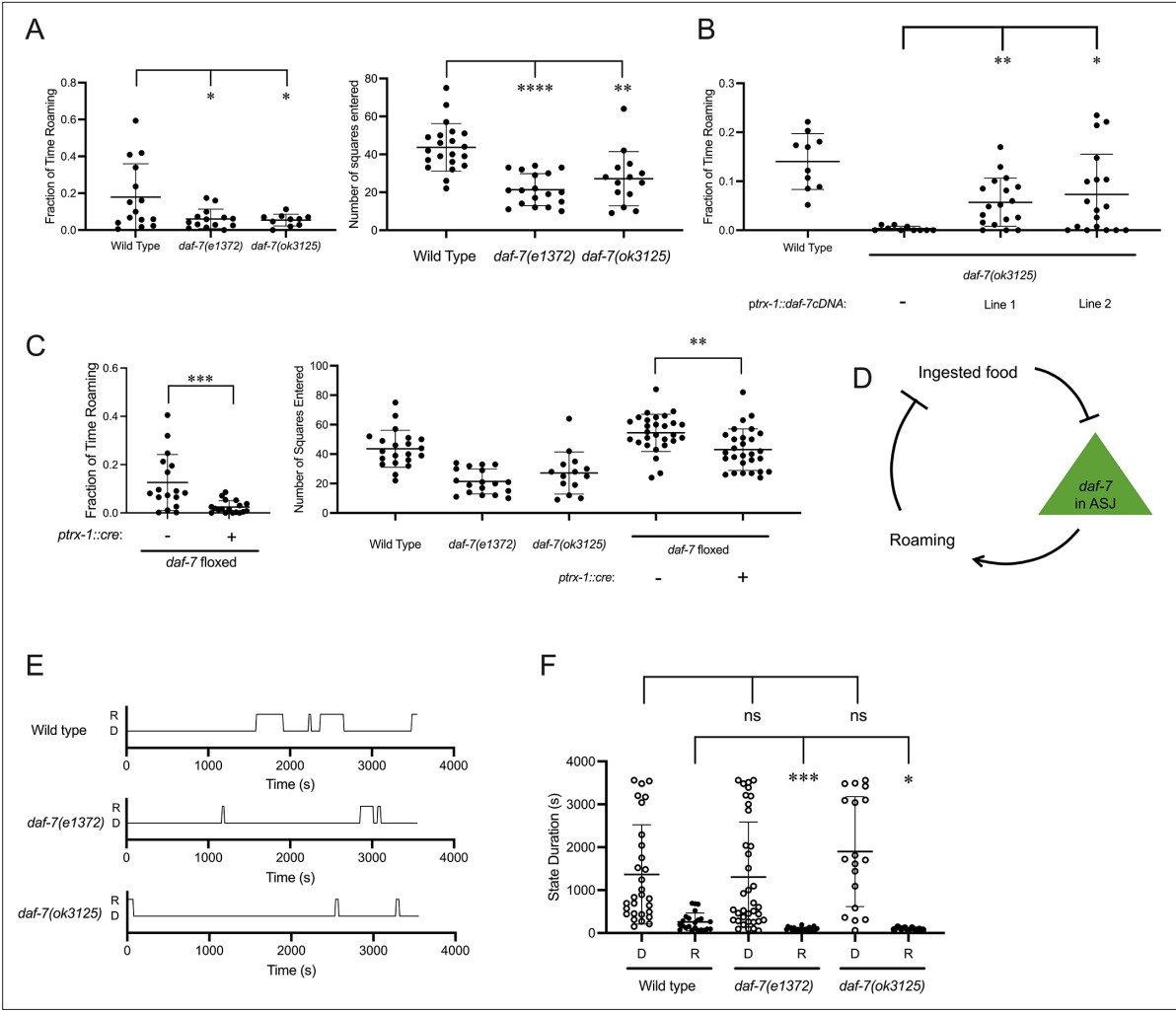

**Figure 2.** *daf-7* expression in the ASJ neurons promotes roaming. (**A**) Fraction of time roaming (left), and number of squares entered in exploration assay (right) of wild-type, *daf-7(e1372)*, and *daf-7(ok3125)* animals. Each point represents an individual animal, and error bars indicate standard deviation. ****p<0.0001, **p<0.01, *p<0.05 as determined by an unpaired two-tailed t-test. (**B**) Fraction of time roaming of wild-type, *daf-7(ok3125)*, and two independent lines where *daf-7* cDNA was expressed under the *trx-1* promoter in a *daf-7(ok3125)* background. Each point represents an individual animal, and error bars indicate standard deviation. **p<0.01, *p<0.05 as determined by an unpaired two-tailed t-test. (**C**) Fraction of time roaming (left), and number of squares entered in exploration assay (right) of wild-type, *daf-7(e1372)*, *daf-7(ok3125)*, and a floxed *daf-7* strain with and without Cre expressed under the *trx-1* promoter. Each point represents an individual animal, and error bars indicate standard deviation. **p<0.01, *p<0.05 as determined by an unpaired two-tailed t-test. (**D**) Model of positive-feedback relationship between food ingestion, *daf-7* expression in the ASJ neurons, and roaming. (**E**) Sample trace files of representative individual wild-type, *daf-7(e1372)*, and *daf-7(ok3125)* animals. R=roaming, D=dwelling. (**F**) Duration of dwelling (open circles) and roaming (closed circles) states for wild-type, *daf-7(e1372)*, and *daf-7(ok3125)* animals. Each point represents a discrete roaming or dwelling period. Error bars indicate standard deviation. ***p<0.001, *p<0.05, ns, not significant as determined by an unpaired two-tailed t-test.

and an exploration assay to measure the general activity levels of animals (*Flavell et al., 2013*, see Materials and Methods), we observed that *daf-7(e1372)* and *daf-7(ok3125)* animals spent a markedly decreased proportion of time in the roaming state compared to wild-type animals, consistent with these prior findings (*Figure 2A*). To determine whether we could detect modulation of roaming and dwelling behaviors caused by changes in *daf-7* expression in the ASJ neurons, we adopted two complementary approaches. First, we found that in a *daf-7(ok3125)* background, rescue of *daf-7* cDNA under the ASJ-specific *trx-1* promoter partially restored the fraction of time animals spent in the roaming state (*Figure 2B*). Second, we examined exploratory behavior in animals that had *daf-7* deleted in the ASJ neurons, using a floxed allele of *daf-7* and Cre expressed under the *trx-1* promoter.

We found that animals with an ASJ-specific *daf-7* deletion explored less of the lawn and spent less time roaming than animals with wild-type expression of *daf-7* (*Figure 2C*).

The relationships between ingested food and *daf-7* expression in the ASJ neurons, *daf-7* expression in the ASJ neurons and roaming, and roaming and amount of food ingested suggest a neuroendocrine gene expression positive-feedback loop that couples the ingestion of bacterial food to the modulation of foraging behavior through *daf-7* expression in the ASJ neurons (*Figure 2D*). When animals are on abundant edible food, the ingestion of this food inhibits *daf-7* expression in the ASJ neurons and attenuates the proportion of time in the roaming state. When animals are removed from edible food but are still exposed to soluble food signals, upregulation of *daf-7* expression in the ASJ neurons promotes an increased proportion of time in the roaming state. Considering that animals can exist in one of the two states while feeding on bacterial food, a decreased fraction of time roaming could be due to shorter roaming states, longer dwelling states, or combination of these factors. We hypothesized that this transcriptional feedback loop might function in the persistence of internal states underlying roaming and dwelling behaviors, so we measured the duration of roaming and dwelling states in animals lacking DAF-7 signaling.

Representative trace files of wild-type, *daf-7(e1372)*, and *daf-7(ok3125)* animals show how DAF-7 can influence roaming and dwelling state duration (*Figure 2E*). Quantification of state duration across multiple animals revealed that *daf-7* animals exhibited shortened roaming states relative to wild-type animals as predicted by the positive feedback loop (*Figure 2F*). We did not observe a difference in dwelling state duration among the genotypes examined, although we suspect the wide variation

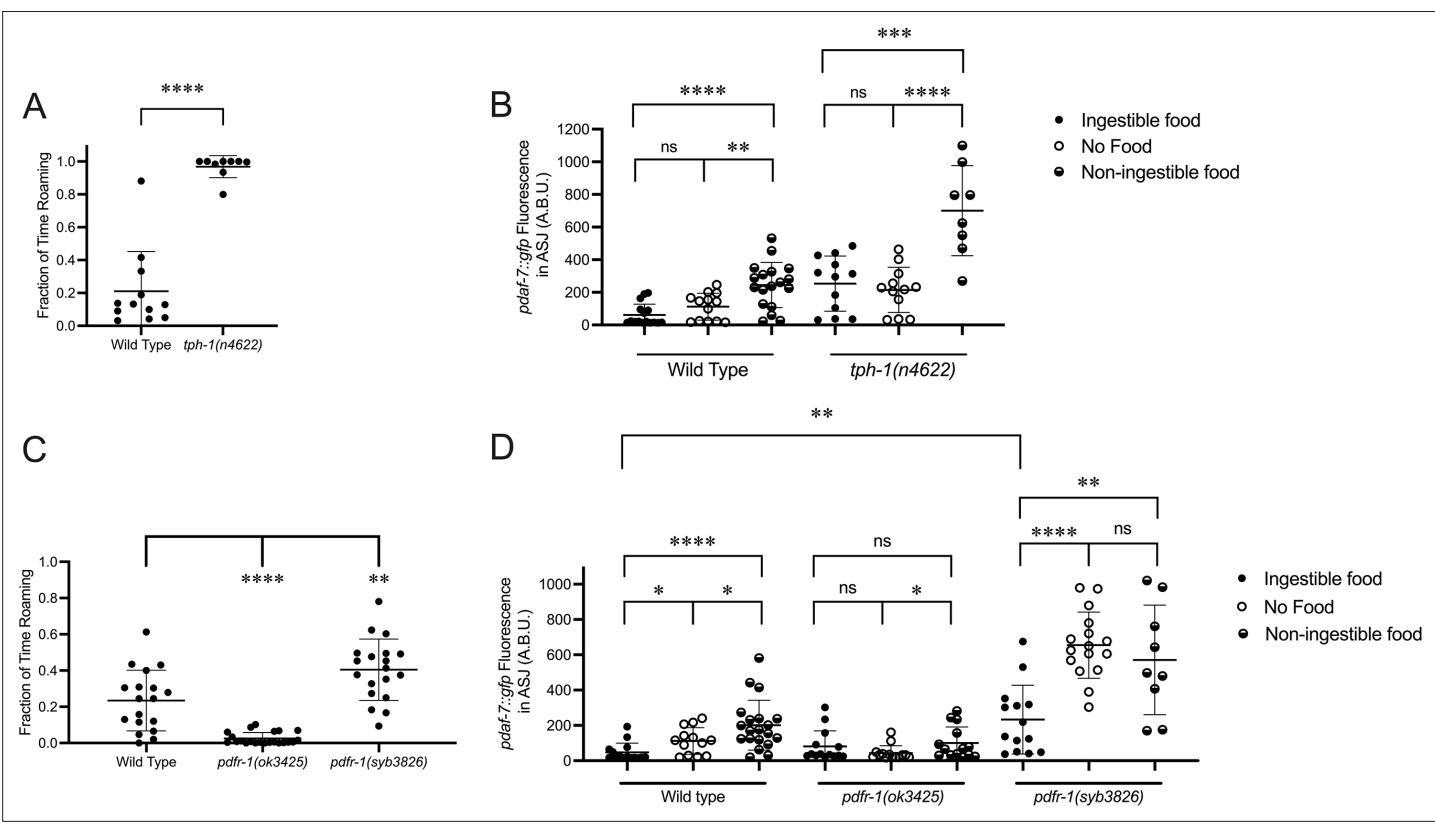

**Figure 3.** Known modulators of roaming and dwelling affect *daf-7* expression in the ASJ neurons. (**A**) Fraction of time roaming of wild-type and *tph-1(n4622)* animals. Each point represents an individual animal, and error bars indicate standard deviation. ****p<0.0001 as determined by an unpaired two-tailed t-test. (**B**) Maximum fluorescence values of *pdaf-7::gfp* in the ASJ neurons of adult wild-type and *tph-1(n4622)* animals under 'Fed,' 'Empty,' and 'Bottom' conditions. Each point represents an individual animal, and error bars indicate standard deviation. ****p<0.0001, ***p<0.001, **p<0.01, ns, not significant as determined by an unpaired two-tailed t-test. (**C**) Fraction of time roaming of wild-type, *pdfr-1(ok3425)*, and *pdfr-1(syb3826)* animals. Each point represents an individual animal, and error bars indicate standard deviation. ****p<0.0001, **p<0.01 as determined by an unpaired two-tailed t-test. (**D**) Maximum fluorescence values of *pdaf-7::gfp* in the ASJ neurons of adult wild-type, *pdfr-1(ok3425)*, and *pdfr-1(syb3826)* animals under 'Fed,' 'Empty,' and 'Bottom' conditions. Each point represents an individual animal, and error bars indicate standard deviation. ****p<0.0001, **p<0.01, *p<0.05, ns, not significant as determined by an unpaired two-tailed t-test.

dwelling state durations exhibited by animals precluded the detection of a statistically significant difference with our analysis.

## Neuromodulatory mechanisms that affect foraging behavior alter *daf-7* expression in the ASJ neurons

The positive-feedback loop supported by our data (*Figure 2D*) suggests a correlation between *daf-7* expression in the ASJ neurons and genetic backgrounds favoring roaming behavior and hence a diminished ingestion of bacterial food. Serotonin has been shown to promote long dwelling states, and thus animals with mutations in *tph-1,* the gene encoding the enzyme that catalyzes the rate-limiting step of serotonin biosynthesis, spend a greater proportion of their time in the roaming state (*Figure 3A*; *Flavell et al., 2013*; *Ji et al., 2021*). Consistent with the correlation between *daf-7* expression in the ASJ neurons and roaming, we see that *tph-1(n4622)* animals constitutively express *daf-7* in the ASJ neurons on ingestible food, no food, and non-ingestible food, with this *daf-7* expression retaining sensitivity to changes in food environments (*Figure 3B*). These results suggest that serotonin signaling may influence *daf-7* expression in parallel to the gustatory and interoceptive food cues.

PDF signaling has also been previously shown to modulate roaming and dwelling behavior, with animals carrying loss-of-function mutations in the PDF receptor gene *pdfr-1* showing a decreased fraction of their time in the roaming state than wild-type animals (*Figure 3C*; *Flavell et al., 2013*; *Ji et al., 2021*). Consistent with the correlation between roaming and *daf-7* expression in ASJ, we observed that animals with loss-of-function mutations in *pdfr-1* do not upregulate *daf-7* transcription in the ASJ neurons on non-ingestible food (*Figure 3D*). From a screen done in parallel to this work, we recovered a putative gain-of-function allele of *pdfr-1*, *pdfr-1(qd385),* carrying a S325F substitution (*Boor, 2022*). Follow-up analysis in an independently generated S325F allele, *pdfr-1(syb3826)*, confirmed that this substitution was responsible for our observed phenotypes. We observed that *pdfr-1(syb3826)* animals exhibited increased roaming behavior compared with wild-type animals (*Figure 3C*) and constitutive expression of *daf-7* in the ASJ neurons in the presence of ingestible food, which was further upregulated under conditions of no bacterial food or non-ingestible food (*Figure 3D*). These results are consistent with a model in which PDFR-1 signaling influences *daf-7* expression in the ASJ neurons by modulating the sensation of the external gustatory food cue.

## SCD-2 controls *daf-7* expression in the ASJ neurons and roaming

To identify additional genetic factors involved in coupling the ingestion of food to *daf-7* expression in the ASJ neurons and its subsequent effect on roaming, we were guided by the isolation of an allele of *hen-1*, *hen-1(qd259),* from a screen we previously performed for genes regulating the expression of *daf-7* from the ASJ neurons in response to *P. aeruginosa* (*Park et al., 2020*). Through genomic rescue experiments, we confirmed that the *hen-1* mutation was the causative lesion in this strain, and animals with mutations in *scd-2*, encoding the receptor of HEN-1, share this phenotype (*Figure 4A*). SCD-2 has characterized for its role in the food-dependent developmental arrest known as dauer diapause and sensory integration (*Reiner et al., 2008*; *Shinkai et al., 2011*; *Wolfe et al., 2019*), and anaplastic lymphoma kinase (ALK), the human ortholog of SCD-2, has been implicated in metabolic phenotypes in humans, mice, and *Drosophila* (*Orthofer et al., 2020*). Taken together, these observations led us to question if SCD-2 might be involved in the regulation of *daf-7* expression in the ASJ neurons in response to changes in food ingestion. We found that animals with loss-of-function alleles of *scd-2* showed reduced upregulation of *daf-7* in the ASJ neurons in response to non-ingestible food compared to wild-type animals, implicating SCD-2 at least partially in this *daf-7* expression response (*Figure 4B*).

To further examine the role of SCD-2 mediating the *daf-7* expression response to ingested food, we generated a gain-of-function allele. Gain-of-function point mutations in ALK are associated with neuroblastoma and most often occur in the kinase domain, with 85% of all ALK point mutations seen at either F1174 or R1275 (*Franco et al., 2013*). F1174L, the most common point mutation, results in ALK autophosphorylation and cytokine-independent growth (*Chen et al., 2008*; *George et al., 2008*; *Hallberg and Palmer, 2013*; *Holla et al., 2017*; *Janoueix-Lerosey et al., 2008*). Analysis of a sequence alignment of SCD-2 and ALK revealed that F1174 in ALK was conserved as F1029 in SCD-2, and we were able to create a gain-of-function allele of *scd-2* by engineering a F1029L substitution (*scd-2(syb2455)*) (*Figure 4C*; *Boor, 2022*). To infer whether SCD-2 activity was sufficient to induce

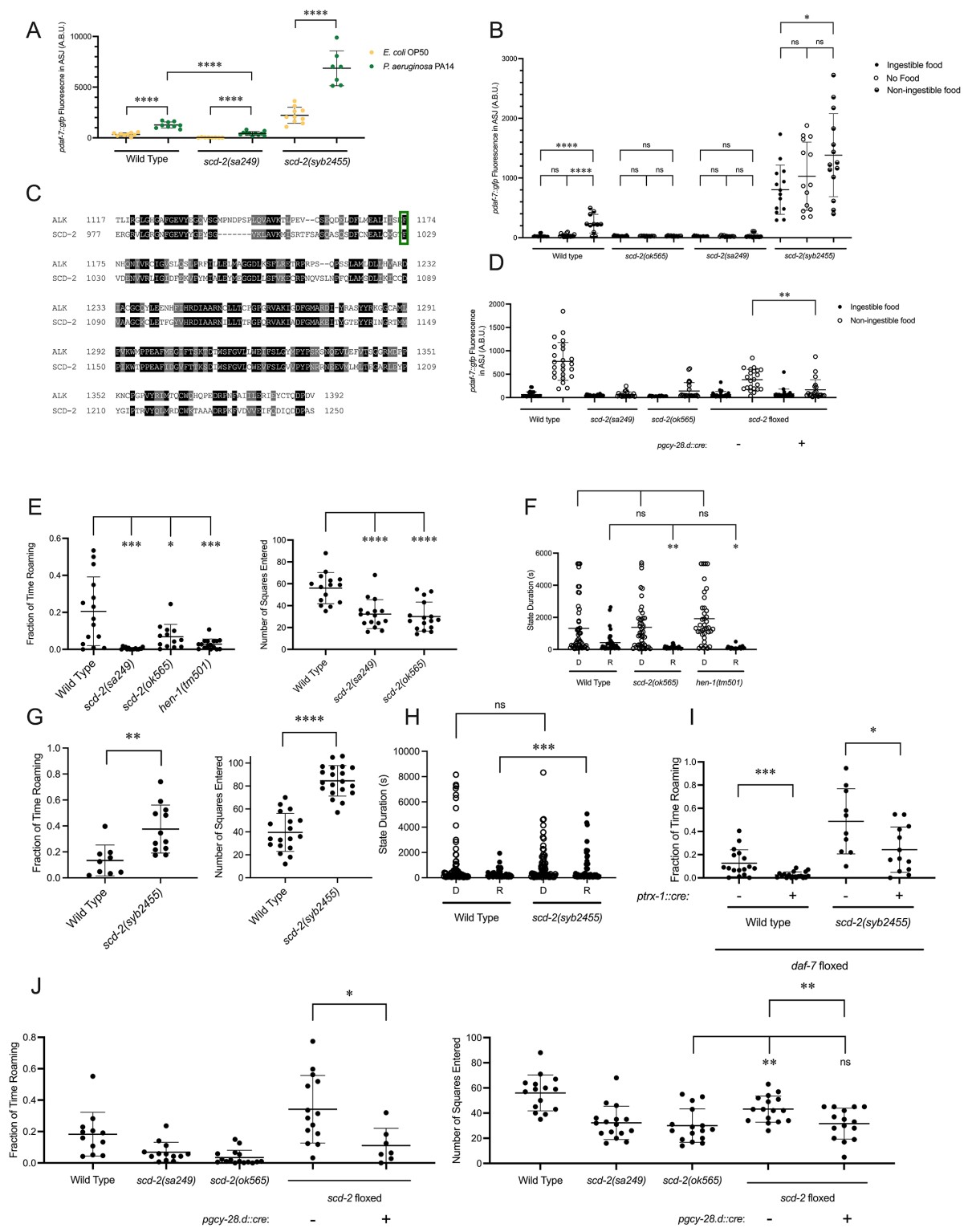

**Figure 4.** SCD-2 regulates *daf-7* expression in the ASJ neurons and roaming behavior. (**A**) Maximum fluorescence values of *pdaf-7::gfp* in the ASJ neurons of adult wild-type, *scd-2(sa249)*, and *scd-2(syb2455)* animals exposed to *E. coli* OP50 and *P. aeruginosa* PA14. Each point represents an individual animal, and error bars indicate standard deviation. ****p<0.0001, as determined by an unpaired two-tailed t-test. (**B**) Maximum fluorescence values of *pdaf-7::gfp* in the ASJ neurons of adult wild-type, *scd-2(ok565), scd-2(sa249),* and *scd-2(syb2455)* animals under 'Fed,' 'Empty,' and 'Bottom' conditions. Each point represents an individual animal, and error bars indicate standard deviation. ****p<0.0001, *p<0.05, ns, not significant as determined by an unpaired two-tailed t-test. (**C**) Sequence alignment of the kinase domains of human anaplastic lymphoma kinase (ALK) (top) and *C.*

*Figure 4 continued on next page*

*Figure 4 continued*

*elegans* SCD-2 (bottom). Amino acids highlighted in black are identical and those in gray are similar. F1174/F1029 is outlined in green. (**D**) Maximum fluorescence values of *pdaf-7::gfp* in the ASJ neurons of adult wild-type, *scd-2(sa249)*, *scd-2(ok565)*, and floxed *scd-2* animals without and with a transgene expressing Cre under the AIA-specific *gcy-28.d* promoter under 'Fed' and 'Bottom' conditions. Each point represents an individual animal, and error bars indicate standard deviation. **$p<0.01$ as determined by an unpaired two-tailed t-test. (**E**) Fraction of time spent roaming (left) and number of squares entered in exploration assay (right) for wild-type, *scd-2(sa249)*, *scd-2(ok565)*, and *hen-1(tm501)* animals. Each point represents an individual animal. Error bars indicate standard deviation. ****$p<0.0001$, ***$p<0.001$, *$p<0.05$ as determined by an unpaired two-tailed t-test. (**F**) Duration of roaming (closed circles) and dwelling (open circles) states for wild-type, *scd-2(ok565)*, and *hen-1(tm501)*. Each point represents a discrete roaming or dwelling period. Error bars indicate standard deviation. **$p<0.01$, *$p<0.05$, ns, not significant as determined by an unpaired two-tailed t-test. (**G**) Fraction of time spent roaming (left) and number of squares entered in exploration assay (right) for wild-type and *scd-2(syb2455)* animals. Each point represents an individual animal. Error bars indicate standard deviation. ****$p<0.0001$, **$p<0.01$ as determined by an unpaired two-tailed t-test. (**H**) Duration of roaming (closed circles) and dwelling (open circles) states for wild-type and *scd-2(syb2455)* animals. Each point represents a discrete roaming or dwelling period. Error bars indicate standard deviation. ***$p<0.001$, ns, not significant as determined by an unpaired two-tailed t-test. (**I**) Fraction of time roaming for wild-type of *scd-2(syb2455)* animals with a floxed allele of *daf-7* without or with an ASJ-specific cre transgene. ***$p<0.001$, *$p<0.05$ as determined by an unpaired two-tailed t-test. (**J**) Left: Fraction of time spent roaming in worm tracker assay for *scd-2(sa249)*, *scd-2(ok565)*, and floxed *scd-2* without or with a transgene expressing Cre under the AIA-specific *gcy-28.d* promoter. Right: number of squares entered in exploration assay for wild-type, *scd-2(sa249)*, *scd-2(ok565)*, and floxed *scd-2* with or without a transgene expressing Cre under the AIA-specific *gcy-28.d* promoter. Each point represents an individual animal. Error bars indicate standard deviation. **$p<0.01$, *$p<0.05$, ns, not significant as determined by an unpaired two-tailed t-test.

expression of *daf-7* in the ASJ neurons in the absence of ingestible bacterial food, we examined the gain-of-function *scd-2(syb2455)* mutant and observed that these animals constitutively expressed *daf-7* in the ASJ neurons even in the presence of ingestible food (*Figure 4B*). We further observed that the magnitude of upregulation of *daf-7* expression in the ASJ neurons when animals were moved from ingestible food to non-ingestible food was reduced in *scd-2(syb2455)* to levels only about one-fifth of that seen in wild-type animals (the ratio of wild-type *daf-7* expression in the ASJ neurons on non-ingestible food to ingestible food = 8.1; the ratio of *scd-2(syb2455) daf-7* expression in the ASJ neurons on non-ingestible food to ingestible food = 1.7) (*Figure 4B*). In contrast to the reduced upregulation seen in the *scd-2(syb2455)* gain-of-function animals when exposed to non-ingestible food, *scd-2(syb2455)* exhibited robust upregulation of *daf-7* expression in the ASJ neurons when exposed to *P. aeruginosa* (*Figure 4A*). The observation that this PA14-dependent upregulation is intact in *scd-2(syb2455)* animals while the non-ingestible food-dependent upregulation is attenuated supports a direct role for SCD-2 in response to ingested food rather than a global control of *daf-7* expression in the ASJ neurons.

Expression of *scd-2* cDNA under the AIA-specific *gcy-28.d* promoter has been shown to be sufficient to rescue behavioral phenotypes of *scd-2* mutants that have been attributed to a role for AIA-expressed SCD-2 in sensory integration (*Shinkai et al., 2011*; *Wolfe et al., 2019*). To ask whether SCD-2 could be functioning in the AIA neurons to regulate *daf-7* expression in the ASJ neurons, we generated a strain with the coding sequence of SCD-2 floxed and introduced a transgene expressing Cre under the *gcy-28.d* promoter. Compared to animals containing a floxed *scd-2* allele without Cre, animals with AIA-specific Cre expression resulting in AIA-specific deletion of *scd-2* showed reduced upregulation of *daf-7* expression in the ASJ neurons in response to non-ingestible food, suggesting that SCD-2 activity in the AIA neurons acts cell-non-autonomously to control *daf-7* expression in the ASJ neurons (*Figure 4D*).

Our finding that SCD-2 regulates the *daf-7* expression in response to ingested food predicts that SCD-2 would also promote roaming behavior. Consistent with this hypothesis, we observed that animals with loss-of-function mutations in *scd-2* spent a lower proportion of time roaming and explored less of the lawn in an exploration assay than wild-type animals (*Figure 4E*). Animals with a loss-of-function allele of *hen-1* also roamed less than wild type (*Figure 4E*). Furthermore, as predicted by the regulation of *daf-7* expression in the ASJ neurons, *scd-2* and *hen-1* animals exhibit shorter roaming states and unchanged dwelling state durations compared to wild-type animals (*Figure 4F*). When we examined the roaming and dwelling behavior of *scd-2(syb2455)* gain-of-function animals, we found that these mutants showed increased roaming behavior and lawn exploration as compared to wild-type animals and exhibited longer roaming states (*Figure 4G and H*). Consistent with SCD-2 mediating roaming partially through its influence on *daf-7* expression in the ASJ neurons, we saw that the increased roaming exhibited by *scd-2(syb2455)* animals was reduced in animals with a

Cre-lox-mediated ASJ-specific *daf-7* deletion (**Figure 4I**). As with the regulation of *daf-7* expression in the ASJ neurons, we observed that Cre-lox-mediated deletion of *scd-2* specifically in the AIA neurons reduced both roaming and exploration compared to floxed *scd-2* animals without Cre, suggesting that SCD-2 functions in the AIA interneurons to promote roaming behavior (**Figure 4J**).

## A neuronal gene expression correlate of internal state dynamics driving foraging behavior

Considering our positive feedback model, we reasoned that conditions under which we have observed *daf-7* expression in the ASJ neurons would correlate with increased roaming behavior. Previous work from our lab has shown that upon reaching reproductive maturity, male *C. elegans* induce *daf-7* expression in their ASJ neurons, which contributes to male mate-searching behavior (**Figure 5A**; *Hilbert and Kim, 2017*). Consistent with the behaviors that have been reported for males and hermaphrodites in bacterial food-leaving assays (*Lipton et al., 2004*), we observed that wild-type males spend a greater proportion of their time in the roaming state than hermaphrodites (**Figure 5B**). We also observed that the *daf-7* expression in the ASJ neurons in males was sensitive to the inhibitory effects of ingested food, although the magnitude of the fold increase in *daf-7* expression from the ASJ neurons was about half that observed for hermaphrodites (**Figure 5A**).

We have also previously reported an upregulation in *daf-7* expression in the ASJ neurons when animals are on pathogenic *P. aeruginosa* PA14 in response to specific secondary metabolites phenazine-1-carboxamide and pyochelin (**Figure 5C**; *Meisel et al., 2014*). Analysis of roaming and dwelling on PA14 revealed that animals significantly increase the fraction of their time in the roaming state on PA14 compared to nonpathogenic *E. coli* OP50 (**Figure 5D**). We observed that upon exposure to the nonpathogenic PA14 mutant ΔgacA, which results in an intermediate level of *daf-7* expression in the ASJ neurons (**Figure 5C**; *Meisel et al., 2014*), animals roamed an intermediate amount between animals on OP50 and pathogenic PA14 (**Figure 5D**).

## Discussion
### A neuroendocrine gene expression feedback loop couples the ingestion of bacterial food to foraging behavior

The data we present here support a model in which bacterial food ingestion regulates roaming and dwelling behavior in part via SCD-2-dependent *daf-7* expression in the ASJ neurons. Our identification of the AIA neurons as the site of SCD-2 action is consistent with a role for the AIA neurons in relaying food signals in other contexts. AIA has been characterized as a downstream synaptic target of multiple food sensory neurons to integrate information about the food environment and influence chemoreceptor expression in other neurons (*Dobosiewicz et al., 2019*; *McLachlan et al., 2022*).

We have observed that the expression of *daf-7* in the ASJ neurons responds to two opposing food cues. An interoceptive cue derived from the sensation of ingested food in the pharynx inhibits the expression of *daf-7* in the ASJ neurons, while the presence of a diffusible non-ingested food cue induces the expression of *daf-7* in the ASJ neurons (**Figure 1B**). In the presence of both food cues, as on abundant ingestible food, the interoceptive food signal is epistatic to the external food signal, promoting exploitation of the current food environment with low levels of *daf-7* expression in the ASJ neurons. Our identification of two opposing food cues is consistent with findings from Wexler et al., which reported that aztreonam-treated bacteria could restore *daf-7* expression in the ASJ neurons of starved males, indicative of an external food cue that is upregulating *daf-7* expression in the ASJ neurons (*Wexler et al., 2020*). Furthermore, the activity of the ASJ neurons have been demonstrated to be modulated by removal of bacterial supernatant (*Zaslaver et al., 2015*), consistent with ASJ responsiveness to an external food cue.

As discussed above, the relationship between foraging behavior, food ingestion, and SCD-2-dependent *daf-7* expression has led us to posit a positive-feedback loop that couples food ingestion with foraging behavior (**Figure 2D**). As animals in a patchy food environment encounter a region with less edible food, our model predicts that a reduction of ingested food would result in *daf-7* expression induction in the ASJ neurons via SCD-2. This increase in *daf-7* expression is correlated with an internal state that favors roaming. As animals roam, they ingest less bacteria, further promoting SCD-2-dependent *daf-7* expression in the ASJ neurons, stabilizing a roaming-prone state. Once a roaming

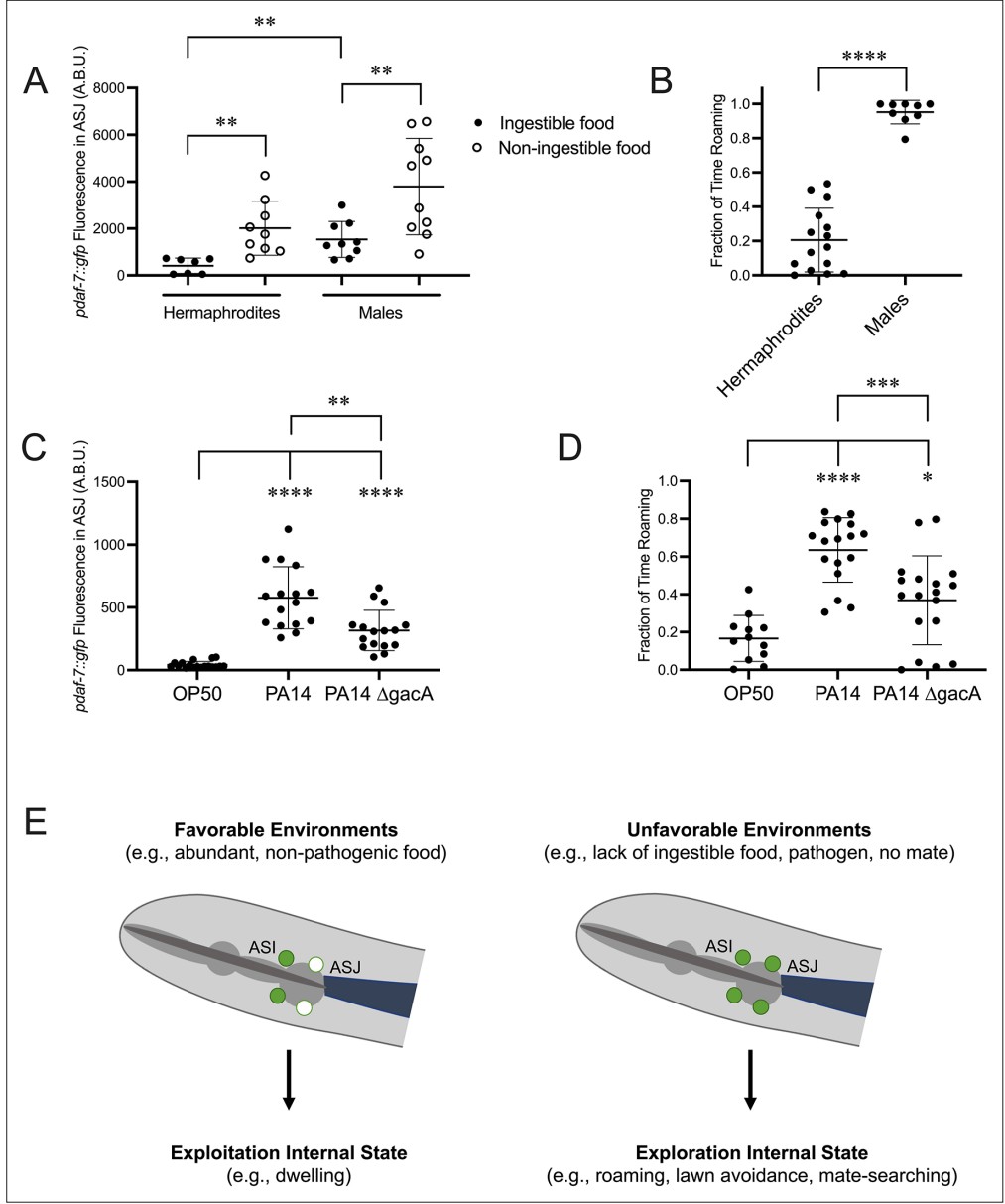

**Figure 5.** DAF-7 expression in the ASJ neurons correlates with an increase in roaming behavior under various conditions. (**A**) Maximum fluorescence values of *pdaf-7::gfp* in the ASJ neurons of adult hermaphrodites and males under 'Fed' and 'Aztreonam' conditions. Each point represents an individual animal, and error bars indicate standard deviation. **p<0.01 as determined by an unpaired t-test. (**B**) Fraction of time roaming of wild-type hermaphrodites and males. Each point represents an individual animal, and error bars indicate standard deviation. ****p<0.0001 as determined by an unpaired t-test. (**C**) Maximum fluorescence values of *pdaf-7::gfp* in the ASJ neurons of adult hermaphrodites fed *E. coli* OP50, *P. aeruginosa* PA14, or *P. aeruginosa* PA14 ΔgacA. Each point represents an individual animal, and error bars indicate standard deviation. ****p<0.0001, **p<0.01 as determined by an unpaired t-test. (**D**) Fraction of time roaming of wild-type hermaphrodites on *E. coli* OP50, *P. aeruginosa* PA14, or *P. aeruginosa* PA14 ΔgacA. Each point represents an individual animal, and error bars indicate standard deviation. ****p<0.0001, ***p<0.001, *p<0.05 as determined by an unpaired t-test. (**E**) *daf-7* expression in the ASJ neurons responds to environmental conditions and is correlated with internal state. Under favorable conditions, *daf-7* is not expressed in the ASJ neurons, and this is correlated with an internal state that favors exploitation of the animal's current environment. Under unfavorable conditions, *daf-7* expression is induced in the ASJ neurons, consistent with an internal state that favors exploration.

animal encounters edible food and begins to eat, the interoceptive sensation of food in the pharynx results in a downregulation of *daf-7* expression in the ASJ neurons, promoting a dwelling state in which the animal can continue to eat. As additional food is ingested, this further reduces *daf-7* expression in the ASJ neurons, stabilizing this internal state that favors dwelling. Moreover, a recent study has highlighted the role of the ASJ neurons in promoting roaming and food-leaving behavior (*Scheer and Bargmann, 2023*). We additionally find that PDF and serotonin signaling appear to contribute to *daf-7* expression in the ASJ neurons in addition to their previously characterized roles in roaming and dwelling (*Flavell et al., 2013*; *Ji et al., 2021*).

A number of recent studies in *C. elegans* point to a key role for neuronal transcriptional responses in the regulation of behavioral plasticity in response to changing environmental stimuli such as food and temperature (*Harris et al., 2023*; *Kyani-Rogers et al., 2022*; *Ryan et al., 2014*; *Wexler et al., 2020*). For example, differential expression of the diacetyl-sensing chemoreceptor ODR-10 in the AWA neurons in response to changing food conditions, which functions downstream of *daf-7* expression in the ASJ neurons of males, has been shown to modulate food-leaving mate-searching behavior (*Ryan et al., 2014*; *Wexler et al., 2020*). We anticipate that the characterization of neuron-specific transcriptional changes that are correlated with *daf-7* expression in the ASJ neurons may yield additional mechanistic insights into how changing environmental and endogenous ingested bacterial food levels modulate internal states driving foraging behavior.

## SCD-2/ALK controls a neuroendocrine response to the ingestion of bacterial food

Here, we show that *daf-7* expression in the ASJ neurons is regulated by the receptor tyrosine kinase SCD-2. SCD-2 and its ligand, HEN-1, have been implicated sensory integration, associative learning, and memory (*Ishihara et al., 2002*; *Shinkai et al., 2011*; *Wolfe et al., 2019*), and SCD-2 was initially characterized for its role in regulating dauer formation, a developmentally arrested state induced by stressful growth conditions, including lack of nutritious food (*Golden and Riddle, 1982*; *Golden and Riddle, 1984b*; *Golden and Riddle, 1984c*; *Golden and Riddle, 1984a*; *Inoue and Thomas, 2000*; *Reiner et al., 2008*). SCD-2 is the *C. elegans* ortholog of ALK. In humans, *ALK* is expressed primarily in the nervous system and influences cell proliferation, differentiation, and survival in response to external stimuli (*Iwahara et al., 1997*). Genetic translocations resulting in gene fusions are common in *ALK*-dependent human cancers, including non-small cell lung cancer, diffuse large B cell lymphoma, squamous cell carcinoma, and renal cell carcinoma (*Hallberg and Palmer, 2013*; *Holla et al., 2017*; *Morris et al., 1994*). Recent studies have suggested a physiological role for ALK in the neuronal control of metabolism. A genome-wide association study looking for genetic variants associated with thinness identified a variant in the first intron of *ALK* (*Orthofer et al., 2020*). Additional studies in *Drosophila* and mice have further implicated ALK in triglyceride accumulation, starvation survival, and metabolic response to high-fat diets (*Cheng et al., 2011*; *Orthofer et al., 2020*; *Woodling et al., 2020*).

Our work describes a role for SCD-2 in regulating foraging behavior and neuroendocrine gene expression in response to changing food conditions. We and others have observed that animals with loss-of-function mutations in *scd-2* behave as if they are in the constant presence of ingested 'good' food; *scd-2* animals dwell more, reduce upregulation of *daf-7* expression in the ASJ neurons in the absence of ingested food, and fail to enter dauer under poor food conditions (*Reiner et al., 2008*). In contrast, animals with the gain-of-function *scd-2(syb2455)* allele act like animals exposed to 'bad' food conditions, as evidenced by increased roaming, constitutive expression of *daf-7* in their ASJ neurons, and food-independent constitutive dauer entry (*Boor, 2022*). If SCD-2 is functioning in responding to changes in food, this may alter conclusions of several previous studies implicating HEN-1 and SCD-2 in sensory integration, as several of the assays employed in these studies have been shown to be affected by the animals' nutritional status (*Ishihara et al., 2002*; *Shinkai et al., 2011*; *Wolfe et al., 2019*). Further investigation into the role of SCD-2 in responding to food vs. sensory integration in these assays could be informative.

Furthermore, the role for SCD-2 in regulating the physiological response to ingested food in *C. elegans* is consistent with the growing body of work tying ALK and its orthologs to thinness and metabolic phenotypes in humans, mice, and *Drosophila* (*Cheng et al., 2011*; *Orthofer et al., 2020*; *Woodling et al., 2020*). Our observations of *scd-2 C. elegans* are consistent with phenotypes

seen in *ALK*$^{-/-}$ mice and *Alk* RNAi *Drosophila*, including increased energy expenditure and reduced triglyceride accumulation (***Orthofer et al., 2020***). Since the phenotypes of *scd-2* animals are associated with abundant food in *C. elegans,* we might expect these *ALK*$^{-/-}$ mice and *Alk* RNAi *Drosophila* to mimic animals that are constantly eating and engage in compensatory mechanisms of increased energy expenditure and triglyceride metabolism. Further characterization of the mechanisms by which SCD-2 and ALK regulate food-dependent behavior, gene expression, and metabolism could reveal new factors governing body weight and have implications in the treatment and prevention of obesity.

### *daf-7* expression in the ASJ neurons is correlated with an internal state favoring exploration

The response of *daf-7* expression in the ASJ neurons to changes in food conditions and its effect on foraging behavior provides an example of the influence of gene expression on internal states. We have identified a transcriptional switch of one gene in two cells that can influence foraging behavior in a manner as equally simple as but on a longer timescale than neuronal firing (***Figure 5E***). This *daf-7*-expressing, pro-roaming internal state might drive other exploratory behaviors that our lab and others have observed under conditions were *daf-7* is expressed in the ASJ neurons. Male *C. elegans* have been observed to upregulate the expression of *daf-7* in their ASJ neurons upon reaching reproductive maturity, and this *daf-7* expression has been implicated in driving mate-searching behavior (***Hilbert and Kim, 2017***). Males must find a hermaphrodite mate to reproduce – a requirement not shared by hermaphrodites – and they often must leave food to do so. The constitutive expression of *daf-7* in the ASJ neurons of adult males could be due to a reduced sensitivity to ingested food allowing for prioritization of searching for a mate over feeding. This is consistent with our observation that *daf-7* expression in the ASJ neurons is upregulated less in response to non-ingestible food in males than in hermaphrodites.

Similarly, our lab has previously reported that two secondary metabolites of pathogenic *P. aeruginosa* can result in the upregulation of *daf-7* expression in the ASJ neurons, which contributes to pathogen avoidance behavior (***Meisel et al., 2014***). The results presented here suggest that perhaps these secondary metabolites are tapping into this foraging circuit and overriding other inputs that regulate *daf-7* expression in the ASJ neurons to promote the animals moving away from the pathogen in a manner like roaming animals seeking nutritious food. In the natural environment of *C. elegans*, pathogenic and nutritious bacteria coexist, and animals must employ mechanisms to avoid infection while still obtaining adequate nutrition. Low levels of nutritious food, the presence of pathogen, or an absence of mates each represents a suboptimal environment and may induce a common internal state that promotes exploration and enhances the chances of encountering more favorable conditions.

Multiple integrative neuronal mechanisms likely converge to establish 'internal state' that modulates organism behavior. Our data reveal that gene expression, specifically the expression of a single gene, *daf-7*, from just two neurons, the ASJ chemosensory neurons, can not only contribute to internal state driving foraging behavior through its role in a neuroendocrine feedback loop, but also serve as a readily detected correlate of internal state underlying foraging behavior.

## Materials and methods

### *C. elegans* strains

*C. elegans* was maintained on *E. coli* OP50 as previously described (***Brenner, 1974***). Daf-c strains were grown at 16°C. See ***Table 1*** for a complete list of strains used in this study.

### Design of *scd-2* gain-of-function allele

We performed a protein alignment of *C. elegans* SCD-2 and human ALK in NCBI BLAST (***Figure 4C***). Using a list of known oncogenic ALK mutations (***Holla et al., 2017***), we screened these residues for conservation or similarity between the *C. elegans* and human protein sequences. Genome editing was done by SunyBiotech using CRISPR technology. Alleles were evaluated in a dauer assay for a gain-of-function Daf-c phenotype, and validated in trans-heterozygote analysis with *scd-2(sa249)* for a dominant phenotype (***Boor, 2022***).

### Preparation of food condition plates

Unless otherwise indicated, all assays were performed on NGM plates with no peptone. Aztreonam-treated bacteria was prepared as previously reported (***Gruninger et al., 2008***), and 40 μL

**Table 1.** Complete list of *C. elegans* strains used in this study.

| Strain Name | Genotype | Source |
| --- | --- | --- |
| N2 | Wild type | *Caenorhabditis* Genetics Center (CGC) |
| JT249 | scd-2(sa249) | CGC |
| RB783 | scd-2(ok565) | CGC |
| JC2154 | hen-1(tm501) | CGC |
| PHX2455 | scd-2(syb2455) | This study/SunyBiotech |
| FK181 | ksIs2[pdaf-7::gfp; rol-6(su1006)] | CGC |
| ZD2540 | ksIs2; scd-2(sa249) | This study |
| ZD930 | ksIs2; scd-2(ok565) | This study |
| ZD918 | ksIs2; hen-1(tm501) | This study |
| ZD2605 | ksIs2; scd-2(syb2455) | This study |
| CB1372 | daf-7(e1372) | CGC |
| ZD715 | daf-7(ok3125) | *Meisel et al., 2014* |
| ZD695 | daf-7(ok3125);qdEx34[ptrx-1::daf-7;pges-1::GFP] | *Meisel et al., 2014* |
| ZD696 | daf-7(ok3125);qdEx35[ptrx-1::daf-7;pges-1::GFP] | *Meisel et al., 2014* |
| ZD2632 | ksIs2; del-3(ok2613);del-7(ok1187) | This study |
| MT14984 | tph-1(n4622) | Horvitz Lab |
| ZD667 | ksIs2; tph-1(n4622) | This study |
| ZD2079 | pdfr-1(ok3425) | *Hilbert and Kim, 2018* |
| PHX3826 | pdfr-1(syb3826) | This study/SunyBiotech |
| ZD1987 | ksIs2; pdfr-1(ok3425); him-5(e1490) | *Hilbert and Kim, 2018* |
| ZD2633 | ksIs2; pdfr-1(syb3826); him-5(e1490) | This study |
| ZD2721 | scd-2(syb5845 syb6052) Backcrossed x2 | This study |
| ZD2722 | daf-7(syb5855 syb5965) qdEx[ptrx-1::Cre; pofm-1::gfp] Backcrossed x2 | This study |
| ZD2752 | scd-2(syb5845 syb6052); ex[gcy-28.dp::cre; ofm-1p::gfp] | This study |
| ZD2766 | scd-2(syb5845 syb6052); ex[gcy-28.dp::cre; ofm-1p::gfp]; ksIs2 | This study |
| ZD2796 | scd-2(syb2455); daf-7(syb5855 syb5965) qdEx[ptrx-1::Cre; pofm-1::gfp] | This study |
| ZD1005 | him-5(e1390); ksIs2 | *Hilbert and Kim, 2018* |

aztreonam-treated food was added to plates containing 10 µg/mL aztreonam. 'Fed' plates for *daf-7* quantification were seeded with 40 µL OP50 grown overnight in a shaking LB culture at 37°C. 'Bottom' plates were seeded with 250 µL OP50, and the agar was inverted with a spatula immediately prior to adding animals. 'Lid' plates were prepared by pouring a spot of NGM agar on the inside of the lid of the plate and seeding 100 µL of OP50 on to the spot. PA14 was grown as previously described and seeded onto SKA plates, as were *E. coli* OP50 controls for these experiments (*Meisel et al., 2014*).

### *pdaf-7::gfp* quantification assays

Plates of gravid animals were bleached and eggs were dropped onto NGM plates seeded with OP50 and grown at 20°C for 67 hr, unless otherwise noted. On the day of imaging, 15–30 day-1 adult animals were transferred by picking to assay plates, where they were incubated for 5 hr at 20°C. Animals to be imaged were mounted on glass slides with agarose pads and 50 mM sodium azide or 5 mM levamisole. All imaging for pictures were conducted on the Zeiss Axioimager Z1. Quantification of GFP brightness was derived from maximum fluorescence values within the ASJ neurons in FIJI.

## Exploration assay

The exploration assay was performed loosely as previously described with several modifications (*Flavell et al., 2013*). 35 mm NGM plates with no peptone were seeded with 500 µL OP50 grown overnight in LB so that the lawn covered the entire surface of the plate. On the day of the assay, one day-1 adult animal was placed on the plate and allowed to explore for 2 hr before being removed. The plate was then superimposed on a grid of 3.5 mm squares and the number of squares crossed by the *C. elegans* tracks was manually counted.

## Roaming/dwelling assay

Animals were egg-laid and grown to day-1 adults. 10 cm NGM plates without peptone were seeded with 2 mL stationary phase *E. coli* OP50 grown overnight in LB. Assays were performed with ~20 animals inside a 6 cm copper ring placed in the center of the seeded plate. After 1 hr for the animals to adjust to their new environment, videos were recorded at 3.75 frames per second for 1.5–3 hr. Videos were analyzed using MBF Biosciences WormLab software (*WormLab, 2020*).

Measurements of speed and bending angle (midpoint) were averaged over 10 s intervals, and values for each 10 s interval were plotted on a scatter plot of speed (µm/s) vs. bending angle (degrees). Quantification of fraction of time spent roaming or dwelling was done by segregating the points of the scatter plots by a horizontal line whose placement was based on the distribution of points in the control condition for each experiment. Values for this speed cutoff ranged from $y=7$ to $y=12$. Points falling above the line were classified as roaming, and those below the line were classified as dwelling. The fraction of time spent roaming was calculated for each animal based on the speed cutoff. Outliers were removed using a ROUT test with Q=1%. The durations of roaming and dwelling states were determined manually by analyzing average speed values for successive 10 s intervals. Only states lasting longer than five intervals (50 s) were considered.

## Statistics

All statistical analysis was performed using the GraphPad Prism software (GraphPad Prism, RRID: SCR_002798). Statistical tests used are indicated in each figure legend.

## Acknowledgements

We thank Bob Horvitz, Steve Flavell, and the Caenorhabditis Genetics Center, which is funded by the NIH Office of Research Infrastructure Programs (P40 OD010440), for strains, and Cori Bargmann for Cre plasmids. We thank current and past members of the Kim lab for discussions. Finally, we acknowledge financial support from NIH grant R35GM141794.

## Additional information

### Funding

| Funder | Grant reference number | Author |
| --- | --- | --- |
| National Institutes of Health | R35GM141794 | Dennis H Kim |

The funders had no role in study design, data collection and interpretation, or the decision to submit the work for publication.

### Author contributions

Sonia A Boor, Conceptualization, Data curation, Formal analysis, Validation, Investigation, Visualization, Methodology, Writing – original draft, Writing – review and editing; Joshua D Meisel, Investigation, Visualization, Writing – review and editing; Dennis H Kim, Conceptualization, Supervision, Funding acquisition, Visualization, Methodology, Writing – original draft, Writing – review and editing

### Author ORCIDs

Sonia A Boor http://orcid.org/0000-0001-5480-3659
Dennis H Kim http://orcid.org/0000-0002-4109-5152

Reviewer #1 (Public Review): https://doi.org/10.7554/eLife.91120.3.sa1
Reviewer #2 (Public Review): https://doi.org/10.7554/eLife.91120.3.sa2
Reviewer #3 (Public Review): https://doi.org/10.7554/eLife.91120.3.sa3
Author Response https://doi.org/10.7554/eLife.91120.3.sa4

## Additional files

### Supplementary files
• MDAR checklist

### Data availability
Source data for Figures 1D, 1E, 1F, 1G, 1H, 2A, 2B, 2C, 2F, 3A, 3B, 3C, 3D, 4A, 4B, 4D, 4E, 4F, 4G, 4H, 4I, 4J, 5A, 5B, 5C, and 5D have been deposited on Dryad. They can be accessed at https://doi.org/10.5061/dryad.2ngf1vhwn. Genotyping primer sequences can also be accessed at https://doi.org/10.5061/dryad.2ngf1vhwn.

The following dataset was generated:

| Author(s) | Year | Dataset title | Dataset URL | Database and Identifier |
|---|---|---|---|---|
| Boor S, Meisel J, Kim D | 2024 | Data for: Neuroendocrine gene expression coupling of interoceptive bacterial food cues to foraging behavior of *C. elegans* | https://doi.org/10.5061/dryad.2ngf1vhwn | Dryad Digital Repository, 10.5061/dryad.2ngf1vhwn |

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
