## [Editor Report · eLife assessment]

This **important** manuscript focuses on the mechanisms by which food signals and food ingestion modulate animal foraging. The authors provide **convincing** support for the interesting idea that chemosensory and interoceptive signals converge on transcriptional regulation of the TGF-beta ligand DAF-7 in a single pair of *C. elegans* chemosensory neurons (ASJ) to regulate behavior. Their studies implicate a conserved signaling molecule, ALK, in this regulation, suggesting a conserved link between food cues and the neuroendocrine control of foraging behavior.

---

## [Referee Report · Reviewer #1 (Public Review)]

Summary:

Here, Boor et al focus on the regulation of daf-7 transcription in the ASJ chemosensory neurons, which has previously shown to be sensitive to a variety of external and internal signals. Interestingly, they find that soluble (but not volatile) signals released by food activate daf-7 expression in ASJ, but that this is counteracted by signals from the ASIC channels del-3 and del-7, previously shown to detect the ingestion of food in the pharynx. Importantly, the authors find that ASJ-derived daf-7 can promote exploration, suggesting a feedback loop that influences locomotor states to promote feeding behavior. They also implicate signals known to regulate exploratory behavior (the neuropeptide receptor PDFR-1 and the neuromodulator serotonin) in the regulation of daf-7 expression in ASJ. Additionally, they identify a novel role for a pathway previously implicated in *C. elegans* sensory behavior, HEN-1/SCD-2, in the regulation of daf-7 in ASJ, suggesting that the SCD-2 homolog ALK may have a conserved role in feeding and metabolism.

Strengths:

The studies reported here, particularly the quantitation of gene expression and the careful behavioral analysis, are rigorously done and interpreted appropriately. The results suggest that, with respect to food, DAF-7 expression encodes a state of "unmet need" - the availability of nearby food to animals that are not currently eating. This is an interesting finding that reinforces and extends our understanding of the neurobiological significance of this important signaling pathway. The identification of a role for ASJ-derived daf-7 in motor behavior is a valuable advance, as is the finding that SCD-2 acts in the AIA interneurons to influence daf-7 expression in ASJ.

Weaknesses:

A limitation of the work is that some mechanistic relationships between the identified signaling pathways remains unclear, but this provides interesting opportunities for future work. There are some minor concerns about the statistical analysis in the paper, but these are unlikely to affect the authors' interpretation of their results.

---

## [Referee Report · Reviewer #2 (Public Review)]

In this work, Boor and colleagues explored the role of microbial food cues in the regulation of neuroendocrine controlled foraging behavior. Consistent with previous reports, the authors find that C. elegans foraging behavior is regulated by the neuroendocrine TGFβ ligand encoded by daf-7. In addition to its known role in the neuroendocrine/sensory ASI neurons, Boor and colleagues show that daf-7 expression is dynamically regulated in the ASJ sensory neurons by microbial food cues - and that this regulation is important for exploration/exploitation balance during foraging. They identify at least two independent pathways by which microbial cues regulate daf-7 expression in ASJ: a gustatory pathway that promotes daf-7 expression and an opposing interoceptive pathway, also chemosensory in nature but which requires microbial ingestion to inhibit daf-7 expression via ASIC channels, encoded by del-3/del-7. In contrast, the authors show that the conserved PDF neuropeptide signaling pathway likely functions via the gustatory pathway to promote daf-7 expression. They further identify a novel role for the *C. elegans* ALK orthologue encoded by scd-2, which acts in interneurons to regulate daf-7 expression and foraging behavior. These results together imply that distinct cues from microbial food are used to regulate the balance between exploration and exploitation via conserved signaling pathways.

Strengths:

The findings that gustatory and interoceptive inputs into foraging behavior are separable and opposing are novel and interesting, which they have shown most clearly in Figure 1 and Figure 3. These data clarify how these parallel chemosensory pathways can be integrated at the level of daf-7 expression.

It is also clear from their results that removal of the interoceptive cue (via transfer to non-digestible food) results in rapid induction of daf-7::gfp in ASJ - suggesting that this pathway is likely chemosensory and not simply nutritive in nature. They have also shown that daf-7 in ASJ plays an important role in the regulation of foraging behavior.

The role of the hen-1/scd-2 pathway in mediating the effects of ingested food is also compelling and well-interpreted, with a few small caveats, described below. This implies that important elements of this food sensing pathway may be conserved in mammals.

Weaknesses:

Although not a weakness of this work per se, the roles of the 5-HT and hen-1/scd-2 pathway remain a bit unclear, likely reflecting their complex genetic contributions to foraging and daf-7 expression. Future work should clarify how these signals are integrated and whether the integration of these pathways improve exploration/exploitation balance to regulate animal fitness.

---

## [Referee Report · Reviewer #3 (Public Review)]

Summary:

In this interesting study, the authors characterize the mechanisms whereby a *C. elegans* TGF-beta DAF-7 responds to various forms of food cues to regulate foraging.

Building on their previous findings that characterized the functional role of daf-7 in the ASJ sensory neurons in response to a bacterial pathogen and in regulating searching behaviors, the authors of this manuscript show that ingestion of *E. coli* OP50, a common laboratory food for the worms, suppresses ASJ expression of daf-7 and secreted water-soluble cues of OP50 increase it. They further show that the level of daf-7 expression in ASJ is positively associated with a higher level of roaming/exploration. The authors identify that the function of a *C. elegans* ortholog of Anaplastic Lymphoma Kinase in the interneurons AIA regulates ASJ expression of daf-7 in response to food information and the related searching behavior.

Strengths:

The study addresses an important question that appeals to a wide readership. The findings are demonstrated by strong results produced from well designed experiments.

---

## [Author Response]

The following is the authors’ response to the original reviews.

**Public Reviews:**

**Reviewer #1 (Public Review):**
Summary:Here, Boor et al focus on the regulation of daf-7 transcription in the ASJ chemosensory neurons, which has previously been shown to be sensitive to a variety of external and internal signals. Interestingly, they find that soluble (but not volatile) signals released by food activate daf-7 expression in ASJ, but that this is counteracted by signals from the ASIC channels del-3 and del-7, previously shown to detect the ingestion of food in the pharynx. Importantly, the authors find that ASJ-derived daf-7 can promote exploration, suggesting a feedback loop that influences locomotor states to promote feeding behavior. They also implicate signals known to regulate exploratory behavior (the neuropeptide receptor PDFR-1 and the neuromodulator serotonin) in the regulation of daf-7 expression in ASJ. Additionally, they identify a novel role for a pathway previously implicated in *C. elegans* sensory behavior, HEN1/SCD-2, in the regulation of daf-7 in ASJ, suggesting that the SCD-2 homolog ALK may have a conserved role in feeding and metabolism.Strengths:The studies reported here, particularly the quantitation of gene expression and the careful behavioral analysis, are rigorously done and interpreted appropriately. The results suggest that, with respect to food, DAF-7 expression encodes a state of "unmet need" - the availability of nearby food to animals that are not currently eating. This is an interesting finding that reinforces and extends our understanding of the neurobiological significance of this important signaling pathway. The identification of a role for ASJ-derived daf-7 in motor behavior is a valuable advance, as is the finding that SCD-2 acts in the AIA interneurons to influence daf-7 expression in ASJ.

We appreciate the Reviewer 1’s thoughtful assessment of our work and inference that the expression of *daf-7* encodes internal state corresponding to “unmet need.” Based on comments of Reviewer 1 and other reviewers, we have revised the title, abstract, and parts of the discussion to highlight not only the functional contribution of *daf-7* expression in the ASJ neurons to behavioral state, but also the remarkable correlation between gene expression and internal state driving foraging behavior.

Weaknesses:A limitation of the work is that some mechanistic relationships between the identified signaling pathways are not carefully examined, but this provides interesting opportunities for future work.

To enable the reader to begin to infer the relative contributions of the identified signaling pathways to the circuitry coupling distinct bacterial cues to foraging behavior, we have added data for the analysis of DAF-7 expression in the ASJ neurons in the *tph-1* and *pdfr-1* mutants in the complete absence of food. Our current leaning is that multiple pathways, including those we have begun to characterize here, may function in parallel to influence DAF-7 expression and internal state driving foraging behavior. Future work to explore this further is certainly of interest.

A minor weakness concerns the experiment in which daf-7 is conditionally deleted from ASJ. This is an ideal approach for probing the function of daf-7, but these experiments seem to be carried out in the well-fed, on-food condition in which control animals should express little or no daf-7 in ASJ. Thus, the experimental design does not allow an assessment of the role of daf-7 under conditions in which its expression is activated (e.g., in animals exposed to un-ingestible food).

The interpretation of genetic analysis in the complete absence of food is complicated by what we think are multiple parallel pathways that function to strongly promote roaming, as indicated in the prior work of Ben Arous et al. Our observation that the conditional deletion of *daf-7* from the ASJ pair of neurons confers altered roaming behavior on a lawn of bacterial food supports a physiological ongoing role for dynamic *daf-7* expression from the ASJ neurons even in the presence of bacterial food that may contribute to the control of transitions between foraging states and the persistence of roaming and dwelling states.

To demonstrate the functional contribution of DAF-7 expression from the ASJ neuron pair during constitutive expression favoring roaming, we examined the roaming behavior of *scd-2(syb2455)* animals that carry a gain-of-function mutation in *scd-2* that promotes roaming and how the selective deletion of *daf-7* from the ASJ neurons in the *scd-2(syb2455)* genetic background influences roaming behavior. This new experiment supports a model in which DAF-7 expression from the ASJ neurons contributes to the increased roaming behavior exhibited by *scd-2(syb2455)* animals. The new experiment is added as Figure 4I.

An additional minor issue concerns the interpretation of the scd-2 experiments. The authors' findings do support a role for scd-2 signaling in the activation of daf-7 expression by un-ingestible food, but the data also suggest that scd-2 signaling is not essential for this effect, as there is still an effect in scd-2 mutants (Figure 4B).

Considering that most of previous Figure 4B is redundant with previous Figure 4D, we removed previous Figure 4B. Our current Figure 4 has redesignated previous Figure 4D as 4B. We have also added qualification to the text to indicate that other pathways may modulate the *daf-7* expression response to ingested food in parallel to SCD-2 signaling.

**Reviewer #2 (Public Review):**
Summary:In this work, Boor and colleagues explored the role of microbial food cues in the regulation of neuroendocrine-controlled foraging behavior. Consistent with previous reports, the authors find that *C. elegans* foraging behavior is regulated by the neuroendocrine TGFβ ligand encoded by daf-7. In addition to its known role in the neuroendocrine/sensory ASI neurons, Boot and colleagues show that daf-7 expression is dynamically regulated in the ASJ sensory neurons by microbial food cues - and that this regulation is important for exploration/exploitation balance during foraging. They identify at least two independent pathways by which microbial cues regulate daf-7 expression in ASJ: a likely gustatory pathway that promotes daf-7 expression and an opposing interoceptive pathway, also likely chemosensory in nature but which requires microbial ingestion to inhibit daf-7 expression. Two neuroendocrine pathways known to regulate foraging (serotonin and PDF-1) appear to act at least in part via daf-7 induction. They further identify a novel role for the *C. elegans* ALK orthologue encoded by scd-2, which acts in interneurons to regulate daf-7 expression and foraging behavior. These results together imply that distinct cues from microbial food are used to regulate the balance between exploration and exploitation via conserved signaling pathways.Strengths:The findings that gustatory and interoceptive inputs into foraging behavior are separable and opposing are novel and interesting, which they have shown clearly in Figure 1. It is also clear from their results that removal of the interoceptive cue (via transfer to non-digestible food) results in rapid induction of daf-7::gfp in ASJ, and that ASJ plays an important role in the regulation of foraging behavior.

We thank Reviewer 2 for underscoring the modulation of neuroendocrine gene expression in the ASJ neuron pair by distinct gustatory and interoceptive inputs derived from bacterial food that we show in Figure 1.

The role of the hen-1/scd-2 pathway in mediating the effects of ingested food is also compelling and well-interpreted. The use of precise gain-of-function alleles further supports their conclusions. This implies that important elements of this food-sensing pathway may be conserved in mammals.

We thank Reviewer 2 for emphasizing the implications of our study on SCD-2/ALK as well as the generation and use of gain-of-function *scd-2* alleles based on oncogenic mutations in ALK.

Weaknesses:What is less clear to me from the work at this stage is how the gustatory input fits into this picture and to what extent can it be strongly concluded that the daf-7regulating pathways that they have identified (del-3/7, 5-HT, PDFR-1, scd-2) act via the interoceptive pathway as opposed to the gustatory pathway.It follows from the work of the Flavell lab that del-3/7 likely acts via the interoceptive pathway in this context as well but this isn't shown directly - e.g. comparing the effects of aztreonam-treated bacteria and complete food removal to controls. The roles of 5-HT and PDFR-1 are even a bit less clear. Are the authors proposing that these are entirely parallel pathways? This could be explained in better detail.

We have added additional data regarding *daf-7* expression from the ASJ neurons in the complete absence of food in the different mutant backgrounds noted by Reviewer 2. Data regarding *daf-7* expression in the ASJ neurons under three distinct conditions—ingestible bacterial food, non-ingestible bacterial food, and the complete absence of food—enable the pairwise comparison of mutant data that allows for inference regarding the relative contributions of the genes to the interoceptive vs. gustatory pathways. In particular, effects on the interoceptive pathway can be inferred from the comparison of *daf-7* expression on ingestible vs. non-ingestible food, whereas effects on the gustatory pathway can be inferred from the comparison of *daf-7* expression on non-ingestible food vs. the absence of food (newly added).

These additional data are most informative for *del-3; del-7* (Figure 1H), where the added data corroborate a role for these genes in the interoceptive pathway, consistent with the findings of the Flavell lab. Specifically, the observation that *daf-7* expression levels are equivalent between wild-type and *del-3;del-7* animals when there is no ingestible food (either no food or non-ingestible food conditions) suggest that DEL-3 and DEL-7 are functioning specifically to sense ingested food.

For *pdfr-1*, the analysis of the gain-of-function allele suggest that this pathway may have a greater relative effect on the gustatory pathway compared with the interoceptive pathway (Figure 3D). The robust upregulation seen in the *pdfr-1(syb3826)* animals between animals on ingestible and non-ingestible food, suggests that the interoceptive regulation is functional in these mutants, while the lack of upregulation between no-food and non-ingestible-food conditions suggests that the gustatory pathway is affected.

The observations with the 5-HT biosynthesis mutant are most consistent with serotonin signaling affecting *daf-7* expression in the ASJ neurons through a mechanism that is parallel to the gustatory and interoceptive inputs into *daf-7* expression in the ASJ neurons, as *tph-1(n4622)* animals appear to have an elevated baseline expression of *daf-7* in the ASJ neurons while retaining sensitivity to both gustatory and interoceptive food cues (Figure 3B).

The data with *scd-2* are consistent with a role in the epistatic interoceptive pathway, considering the roughly equivalent levels of *daf-7* expression in the ASJ neurons under all food conditions in *scd-2(syb2455)* animals (Figure 4B). However it is difficult to exclude the possibility that SCD-2 functions in both pathways or parallel to the gustatory and interoceptive inputs.

While we agree that our genetic analysis alone cannot distinguish between genes acting in parallel or directly in serial with the gustatory or interoceptive inputs. Our data do establish that signaling through SCD-2, 5-HT or PDFR-1-dependent pathways can act on the same gene expression and signaling node (i.e. *daf-7* expression in the ASJ neurons) to modulate the effects of bacterial food inputs on foraging behavior, with the effects on *daf-7* expression in the ASJ neurons in *scd-2*, *tph-1* and *pdfr-1* mutants correlating with their effects on roaming and dwelling behaviors.

It would also be helpful to elaborate more on why the identified transcriptional positive feedback loop is predicted to extend roaming state duration - as opposed to some other mechanism of increasing roaming such as increased probability of roaming state initiation. This doesn't seem self-evident to me.

Given that animals can exist in only two states, the increased probability of roaming state initiation would present as shorter dwelling states, which we do not see for *daf-7* mutants. As described in Flavell, et al., 2013, a decreased fraction of time roaming can be attributed to longer dwelling states, shorter roaming states, or both. Our positive feedback loop is predicted to extend roaming states because of the predicted effect of DAF-7 on stabilizing the roaming state.

Related to this point is the somewhat confusing conclusion that the effects of tph-1 and pdfr-1 mutations on daf-7 expression are due to changes in ingestion during roaming/dwelling. From my understanding (e.g. Cermak et al., 2020), pharyngeal pumping rate does not reliably decrease during roaming - so is it clear that there are in fact lower rates of ingestion during roaming in their experiments?

This is an interesting point. Despite consistent pumping rates, we still believe that roaming animals ingest less food than dwelling animals. For instance, dwelling animals are localized to areas with bacterial food, while roaming animals might traverse patches with no food where pumping does not result in food ingestion.

If so, why does increased roaming (via tph-1 mutation) result in further increases in daf-7 expression in animals fed aztreonam-treated food (Fig 3B)?

This is possibly because although roaming animals are eating less, when animals are on non-ingestible food, they’re not eating at all, resulting in further *daf-7* upregulation.

Alternatively, there could be a direct signaling connection between the 5-HT/PDFR-1 pathways and daf-7 expression which could be acknowledged or explained.

Yes, this is certainly possible. We do not propose that all of the difference in *daf-7* expression is due to changes in foraging behavior, but rather we are highlighting further instances of the correlation between *daf-7* expression in the ASJ neurons and roaming. For instance, in the case of our *tph-1* mutants, we see a relatively modest effect on *daf-7* expression in the ASJ neurons but a large difference in the fraction of time roaming. This suggests that the magnitude of change in one (*daf-7* expression in ASJ or roaming) does not predict the magnitude of the change in the other, but rather that they trend in the same direction.

**Reviewer #3 (Public Review):**
Summary:In this interesting study, the authors examine the function of a *C. elegans* neuroendocrine TGF-beta ligand DAF-7 in regulating foraging movement in response to signals of food and ingestion. Building on their previous findings that demonstrate the critical role of daf-7 in a sensory neuron ASJ in behavioral response to pathogenic *P. aeruginosa* PA14 bacteria and different foraging behavior between hermaphrodite and male worms, the authors show, here, that ingestion of *E. coli* OP50, a common food for the worms, suppresses ASJ expression of daf-7 and secreted water-soluble cues of OP50 increases it. They further showed that the level of daf-7 expression in ASJ is positively associated with a higher level of roaming/exploration movement.Furthermore, the authors identify that a *C. elegans* ortholog of Anaplastic Lymphoma Kinase, scd-2, functions in an interneuron AIA to regulate ASJ expression of daf-7 in response to food ingestion and related cues. These findings place the DAF-7 TGF-beta ligand in the intersection of environmental food conditions, food intake, and foodsearching behavior to provide insights into how orchestrated neural functions and behaviors are generated under various internal and external conditions.Strengths:The study addresses an important question that appeals to a wide readership. The findings are demonstrated by generally strong results from carefully designed experiments.

We thank Reviewer 3 for the comments and interest in the work.

Weaknesses:However, a few questions remain to provide a complete picture of the regulatory pathways and some analyses need to be strengthened. Specifically,1. The authors show that diffusible cues of bacteria OP50 increase daf-7 expression in ASJ which is suppressed by ingestible food. Their results on del-3 and del-7 suggest that NSM neuron suppresses daf-7 ASJ expression. What sensory neurons respond to bacterial diffusible cues to increase daf-7 expression of ASJ? Since ASJ is able to respond to some bacterial metabolites, does it directly regulate daf-7 expression in response to diffusible cues of OP50 or does it depend on neurotransmission for the regulation? Some level of exploration in this question would provide more insights into the regulatory network of daf-7.

The focus of our study has been on the modulation of *daf-7* expression in the ASJ neurons by distinct bacterial food cues and the downstream neuroendocrine circuitry that is influenced. The question of whether bacterial cues are directly sensed by the ASJ neurons remains unresolved by our study. However, we have previously demonstrated that the *daf-7* expression in the ASJ neurons induced by *P. aeruginosa* metabolites is likely the result of direct detection by the ASJ neurons. We would also note (and have added to the manuscript) the observation of Zaslaver et al. (2015), in which increased calcium transients were observed in the ASJ neurons in response to the withdrawal of *E. coli* OP50 supernatant, which is consistent with our observations of the effect of a soluble bacterial food signal on *daf-7* expression in the ASJ neurons.

1. The results including those in Figure 2 strongly support that daf-7 in ASJ is required for roaming. Meanwhile, authors also observe increased daf-7 expression in ASJ under several conditions, such as non-ingestible food. Does non-ingestible food induce more roaming?

Yes, this has been published by Ben Arous, et al., 2009. Figure 3C shows increased roaming on aztreonam-treated food. We have added specific mention of this in the text.

It would complete the regulatory loop by testing whether a higher (than wild type) level of daf-7 in ASJ could further increase roaming. The results in pdf-1 and scd-2 gain-of-function alleles support more ASJ leads to more roaming, but the effect of these gain-of-function alleles may not be ASJ-specific and it would be interesting to know whether ASJ-specific increase of daf-7 leads to a higher level of roaming. In my opinion, either outcome would be informative and strengthen our understanding of the critical function of daf-7 in ASJ demonstrated here.

We looked at roaming in animals with a *ptrx-1::daf-7 cDNA* transgene in a wild-type background and did not see changes in the fraction of time animals roam. However, multiple experimental factors could contribute to our inability to detect an effect, including relative promoter strength and context of other variables that alter *daf-7* expression. Nevertheless, our data confirmed that ASJ neuron-specific expression of *daf-7* cDNA can increase roaming in a *daf-7* mutant background (Figure 2B).

We have also included an experiment (Figure 4I) looking at roaming in the *scd-2(syb2455)* gain-of-function animals in animals with *daf-7* deleted from the ASJ neurons. These results suggest that part of the increased roaming seen in these *scd-2(syb2455)* animals is specifically due to increased *daf-7* expression in the ASJ neurons.

1. The analyses in Figure 4 cannot fully support "We further observed that the magnitude of upregulation of daf-7 expression in the ASJ neurons when animals were moved from ingestible food to non-ingestible food was reduced in scd-2(syb2455) to levels only about one-fourth of those seen in wild-type animals (Figure 4D)...", because the authors tested and found the difference in daf-7 expression between ingestible and non-ingestible food conditions in both wild type and the mutant worms. The authors did not analyze whether the induction was different between wild type and mutant. Under the ingestible food condition, ASJ expression of daf-7 already looks different in scd-2(syb2455).

We appreciate the reviewer pointing out our lack of clarity in discussing our analysis of the data. The 4x difference represents the difference in fold change from ingested to noningested food in wild type and *scd-2(syb2455*) backgrounds. For wild-type animals, daf-7 expression in the ASJ neurons on non-ingestible food is 8.1-times higher on non-ingestible food than on ingestible food. In *scd-2(syb2455)* animals, this difference is 1.7 times. We have clarified this in the text.

1. The authors used unpaired two-tailed t-tests for all the statistical analyses, including when there are multiple groups of data and more than one treatment. In their previous study Meisel et al 2014, the authors used one-way ANOVA, followed by Dunnett's or Tukey's multiple comparison test when they analyzed daf-7 expression or lawn leaving in different mutants or under different bacterial conditions. It is not clear why a two-tailed t-test was used in similar analyses in this study

We have performed one-way ANOVAs for all comparisons included, and the results were largely consistent with what we found for t-tests. Ultimately, for our analysis we were most interested in pairwise comparisons and decided that t-tests would be most appropriate.

**Reviewer #1 (Recommendations For The Authors):*Line 170: For clarity, I suggest editing this to: "When animals are removed from edible food *but are still exposed to soluble food signals*, upregulation of daf-7..."

We have edited this in the text and appreciate the suggestion.

The authors report that pdfr-1(syb3826) was retrieved from "a screen done in parallel to this work." syb3826 is a Suny Biotech allele, suggesting that this screen may not have been done in the authors' lab but rather outsourced. Some additional details might be useful.

This S325F allele was originally recovered as qd385 in an EMS screen performed in our lab. syb3826 is an independently generated Suny Biotech allele we ordered to confirm that the S325F substitution in PDFR-1 was responsible for our phenotypes. This has been clarified in the text.

Line 210: Please provide a citation for the screen that identified hen-1(qd259).

This is the first time the allele is being published. The screen is included in two theses from our lab, Meisel 2016 and Park 2019.

Line 214: It would be useful here to also mention the previously identified role of scd2 in sensory integration.

Yes, we have added this to the text. Additionally, we have included a couple of sentences in the discussion about how previous studies that have found a role for SCD-2 in sensory integration may instead be detecting the role for SCD-2 in food sensing, as many of the assays used for sensory integration are also sensitive to nutritional status of the animals.

Line 271: Please provide a citation for the sex differences in food-leaving behavior (Lipton 2004 PMID 15329389 is the first careful characterization of this).We have added this to the text.